# Validation of the Modified Helkimo Clinical Index for Diagnosing Temporomandibular Disorders in a Romanian Patient Sample

**DOI:** 10.3390/diagnostics15182347

**Published:** 2025-09-16

**Authors:** Dorin Ioan Cocoș, Sorana Maria Bucur, Mariana Păcurar, Kamel Earar

**Affiliations:** 1Faculty of Dental Medicine, “Dunărea de Jos” University, 800008 Galați, Romania; cdorin1123@gmail.com (D.I.C.); kamel.earar@ugal.ro (K.E.); 2 Department of Dentistry, Faculty of Medicine, Dimitrie Cantemir University of Târgu Mureș, 3-5 Bodoni Sandor Street, 540545 Târgu Mureș, Romania; 3 Department of Orthodontics, Faculty of Dentistry, George Emil Palade University of Medicine, Pharmacy, Science, and Technology of Târgu Mureș, 38 Gheorghe Marinescu Street, 540142 Târgu Mureș, Romania

**Keywords:** temporomandibular joint disorders, diagnosis, reproducibility of results, sensitivity and specificity, Romania

## Abstract

**Background and objectives:** Temporomandibular disorders (TMDs) are common but challenging to diagnose in routine practice due to the complexity of the DC/TMD protocol. The study assesses the diagnostic validity and reliability of the modified Helkimo Index (mHI) in a Romanian patient cohort. **Methods:** A cross-sectional study of 164 participants (82 TMD patients, 82 controls) evaluated the diagnostic performance and reliability of the mHI against the DC/TMD protocol. **Results**: The mHI demonstrated high diagnostic accuracy (sensitivity 86%, specificity 84%, AUC = 0.89) and strong agreement with DC/TMD diagnoses (r = 0.83, *p* < 0.001). Reliability was excellent (inter-examiner ICC = 0.87; intra-examiner ICC = 0.91). Application time was significantly shorter (5–10 min) than the DC/TMD protocol (16–20 min). **Conclusions:** The mHI is a valid, reliable, and time-efficient tool for TMD diagnosis, supporting its integration into general dental practice and broader screening initiatives.

## 1. Introduction

Temporomandibular disorders (TMDs) are a heterogeneous group of conditions involving the temporomandibular joint (TMJ), masticatory muscles, and associated structures. They are commonly characterized by orofacial pain, restricted mandibular function, joint sounds, and reduced quality of life [1]. Prevalence estimates range between 10% and 40% worldwide, with higher rates among women and individuals in early to middle adulthood [2,3].

The etiology of TMDs is multifactorial, encompassing biomechanical, traumatic, psychological, and occlusal factors [4,5,6]. Accurate diagnosis is critical to differentiate TMD from other orofacial pain conditions, and the Diagnostic Criteria for Temporomandibular Disorders (DC/TMD) are currently considered the gold standard [7,8,9,10]. Although the DC/TMD protocol is considered the gold standard, its application is time-intensive (approximately 16–20 min) and requires specialized training. These demands limit its feasibility in general dental practice and low-resource settings, motivating the need for simplified diagnostic tools such as the modified Helkimo Index.

Therefore, there is a growing need in clinical and research settings for alternative diagnostic tools that are more time-efficient, user-friendly, and easier to implement [11,12,13].

Simplified indices have been proposed to improve efficiency without compromising diagnostic accuracy [11,12,13]. The Helkimo Index, originally developed in the 1970s [14], has undergone modifications to address concerns regarding validity and reproducibility. The modified Helkimo Index (mHI) has shown promise as a more practical diagnostic tool, yet evidence from validation studies remains inconsistent and scarce, particularly across diverse populations [14,15,16]. Its use has declined significantly in international literature over the past decades due to criticisms regarding construct validity and the lack of standardized calibration methods [14]. Modified versions of this index have addressed several of these limitations, improving both clinical metric consistency and diagnostic accuracy, and thus show promise in settings where the full application of the DC/TMD protocol may be impractical [14,15]. In this context, a clinical re-evaluation of the modified Helkimo Index (mHI) is warranted, especially to support the expansion of efficient and rapid diagnostic options for TMD in general dental practice and population-level studies [15].

The present study provides a rigorous clinical validation of the mHI in a Romanian patient cohort, directly comparing its diagnostic performance and reliability with the DC/TMD protocol. To our knowledge, this is the first study to evaluate the mHI against the DC/TMD in this population while also quantifying application time. By combining accuracy metrics with feasibility assessment, we highlight the potential of the mHI as a simple, reliable, and time-efficient tool for both clinical practice and population-level screening.

## 2. Materials and Methods

This study was designed as a cross-sectional clinical validation for the diagnostic validity and reliability of the modified Helkimo Index (mHI) in temporomandibular disorders (TMD). The Diagnostic Criteria for Temporomandibular Disorders (DC/TMD) served as the gold-standard method for reference comparison. The Numeric Pain Rating Scale (NPRS) was used to evaluate the correlation between clinical findings and patients’ perceived pain intensity.

### 2.1. Ethical Considerations

The study protocol was approved by the Ethics Committee of “Dunărea de Jos” University of Galați, Romania (Approval No. CEU/16, 9 December 2024). All clinical assessments were conducted within the Department of Pediatric and Adult Dentistry at the County Clinical Emergency Hospital in Galați.

Ethical approval was obtained from the Hospital’s Ethics Committee and the University Ethics Board. Before enrollment, all participants provided written informed consent after receiving comprehensive information regarding the study objectives, procedures involved, potential benefits, and possible risks associated with participation.

### 2.2. Participants

164 participants recruited from the Department of Pediatric and Adult Dentistry at the County Clinical Emergency Hospital in Romania between January and April 2025 were included in the study. A priori sample size calculation was conducted to ensure adequate statistical power. Based on an expected effect size of d = 0.5, α = 0.05, and power = 0.80, a minimum of 128 participants was required. Our final sample of 164 (82 per group) exceeded this threshold, ensuring sufficient precision for validity and reliability analyses. Eligible participants were between 18 and 45 years of age.

-The sample was divided into two distinct groups: TMD Group: 82 patients diagnosed with temporomandibular disorders based on the DC/TMD protocol. Diagnoses were established independently by two calibrated examiners who underwent formal training sessions before the study, including consensus-building exercises and supervised pilot cases to ensure standardization and reproducibility.-Control Group: 82 healthy individuals with no clinical signs or symptoms of TMD, as determined through clinical evaluation and a structured screening questionnaire. Controls were age- and sex-matched to the TMD group.

### 2.3. Inclusion and Exclusion Criteria

Inclusion Criteria:-Age ≥ 18 years;-Willingness to participate in all scheduled evaluations;-Provision of written informed consent.

Exclusion Criteria:-History of craniofacial surgery or trauma;-Ongoing orthodontic treatment;-Known TMJ pathologies (e.g., neoplasms);-Severe systemic diseases affecting TMJ function (e.g., rheumatoid arthritis);-Recent use of medications altering TMD perception (e.g., analgesics, muscle relaxants);-Declined consent.

Although bruxism, psychological stress, and depression are recognized contributors to TMD, they were not used as exclusion criteria to preserve ecological validity and reflect the clinical heterogeneity typically encountered in practice. These factors were not directly measured in the present study; however, their potential influence was minimized by (i) matching groups for age and sex, (ii) applying standardized diagnostic protocols, and (iii) using statistical models (GEE) to adjust for confounders. Future studies incorporating validated psychosocial assessments are recommended to explore their interaction with diagnostic outcomes.

### 2.4. Clinical Measurement Methodology

The diagnostic validity of the modified Helkimo Index (mHI) was assessed against the DC/TMD protocol within the framework of a structured Validation Project. Reference diagnoses were established by consensus between two independent TMD specialists. The DC/TMD protocol served as the gold-standard diagnostic method for confirming the presence of temporomandibular disorders (TMD) [7].

The DC/TMD clinical examination protocol includes 12 core items [7,16,17], encompassing: the presence of muscle and joint pain; pain induced during mandibular movements; palpation-evoked pain in the masticatory muscles and temporomandibular joint; occlusal assessment; and the presence of joint sounds such as clicks or crepitus. It also evaluates mandibular movement limitations, including restricted mouth opening, lateral excursions, and protrusion. Any headache that might be associated with mandibular function is also assessed. The final diagnosis is established through a decision tree algorithm that integrates these clinical variables.

Subjective pain perception was evaluated using the Numeric Pain Rating Scale (NPRS), a widely accepted tool for clinical pain assessment. Participants were asked to rate their perceived discomfort on a numerical scale from 0 to 10, where 0 indicates the complete absence of pain and 10 represents the most intense pain imaginable. This approach provides standardized and comparable data regarding the severity of painful symptoms. NPRS is valued for its ease of use, patient comprehensibility, and validated applicability across diverse clinical contexts [18].

Pain scores were interpreted as follows:-0: No pain-1–3: Mild pain-4–6: Moderate pain-7–10: Severe pain

This classification offers clinicians valuable insight into symptom severity and its potential impact on patients’ daily activities [19]. In clinical validation studies such as the present [20], NPRS plays a role in correlating clinical index scores with patients’ subjective pain experiences, contributing to a comprehensive understanding of their clinical status.

NPRS was used to assess pain intensity in two anatomical regions relevant to TMD pathology: the cervical region and the temporomandibular joint (TMJ). The selection of this tool allowed for precise quantification of pain in both areas. Moreover, it facilitated monitoring of symptom progression during the observation period and supported the effectiveness of any therapeutic interventions. This multidimensional approach offered a detailed perspective on the musculoskeletal impact of pain among participants.

The modified Helkimo Index (mHI) represents a clinically updated refinement of the original Helkimo Index introduced by Martti Helkimo in 1974. Building on the initial framework, subsequent revisions, most notably those proposed by Maglione et al., introduced improved scoring scales, refined pain assessment criteria, and standardized reproducibility protocols, addressing earlier concerns regarding construct validity and enhancing diagnostic applicability. As Table 1 and Table 2 outline, the mHI enables structured evaluation of temporomandibular dysfunction severity, offering a more accurate and reliable tool for contemporary clinical and research settings [15]. Compared with the DC/TMD protocol, which remains the gold standard but is resource-intensive and time-consuming, the mHI provides a pragmatic balance between diagnostic precision and operational efficiency, making it particularly suitable for general dental practice and large-scale screening contexts.

The scores obtained by applying the clinical criteria from Table 1 were used to classify the severity of temporomandibular dysfunction, according to the categories presented in Table 2.

### 2.5. Classification of TMD Severity

Table 2 presents a structured classification of TMD severity based on the total score obtained by the modified Helkimo Index (mHI). This scoring system facilitates the clinical interpretation of dysfunction intensity and supports diagnostic decision-making and treatment planning [15].

### 2.6. Statistical Analysis

For continuous variables, Welch’s *t*-test was applied when assumptions of normality and variance equality were met. For non-normally distributed data, the Mann–Whitney U test was used. This dual approach ensured robust handling of different data distributions.

Statistical analyses were conducted using R version 4.3.2, employing the following packages:-tidyverse for data cleaning and manipulation;-stats and psych for inferential testing and effect size estimation;-irr for reliability coefficients;-pROC for ROC curve analysis and AUC comparisons.

Key results were cross-validated using IBM SPSS Statistics version 27.

The normality of continuous variables was assessed using the Shapiro–Wilk test. Normally distributed data are reported as mean ± standard deviation (SD) and compared between groups using the Welch’s *t*-test. Homogeneity of variances was verified using Levene’s test. Non-normally distributed variables are reported as median and interquartile range (IQR) and compared using the Mann–Whitney U test. Categorical frequencies were compared using the Chi-square test (χ^2^) or Fisher’s exact test, depending on expected cell sizes.

Effect sizes were quantified using:-Cohen’s d for continuous parametric comparisons;-r for nonparametric comparisons;-Φ coefficient (phi) for categorical variables.

Inter- and intra-examiner reliability for Helkimo scores was assessed using the Intraclass Correlation Coefficient (ICC), model 2.1, based on absolute agreement, and Cohen’s kappa (κ) for item-level agreement. Confidence intervals (95%) were calculated using bootstrap resampling (1000 iterations). Interpretation thresholds followed the guidelines by Koo and Li [21]:-ICC < 0.50 = poor-0.50–0.75 = moderate-0.75–0.90 = good-0.90 = excellent.

The linear correlation between mHI and DC/TMD scores was evaluated using the Pearson correlation coefficient (r). Correlation strength was interpreted according to Evans, where r ≥ 0.80 denotes a strong association.

Receiver Operating Characteristic (ROC) curves were generated for both diagnostic instruments. Area under the Curve (AUC) values were compared using DeLong’s test. The optimal cutoff point was determined using the Youden Index (J = Sensitivity + Specificity − 1). For the identified thresholds, sensitivity, specificity, positive predictive value (PPV)**,** negative predictive value (NPV), and likelihood ratios (LR+ and LR−) were reported, with exact binomial confidence intervals (Clopper–Pearson method).

Generalized Estimating Equations (GEE) with a logit link and exchangeable correlation structure were applied to account for intra-group dependency and adjust for confounding variables such as age and sex.

The study was conducted in accordance with the STROBE (Strengthening the Reporting of Observational Studies in Epidemiology) guidelines for observational studies and the STARD 2015 (Standards for Reporting Diagnostic Accuracy Studies) checklist. The completed checklists are provided as Appendix A to ensure transparent and standardized reporting.

## 3. Results

A total of 212 individuals were initially contacted for potential inclusion. After applying the eligibility criteria and accounting for refusals, 164 participants were enrolled. Of these, 82 were clinically diagnosed with TMD and 82 were healthy controls. Forty-eight individuals were excluded due to non-eligibility or refusal. Participant demographics are summarized in Table 3.

The mean age was slightly lower in the TMD group (28.0 ± 6.0 years) compared with controls (31.0 ± 6.5 years; *p* = 0.02), with a moderate effect size (d = 0.49). Sex and BMI distributions were comparable across groups (*p* > 0.3). Interestingly, a significantly higher proportion of TMD patients reported being athletes (60% vs. 48%; *p* = 0.004), suggesting a possible association between sports activity and TMD occurrence. Urban–rural distribution also differed significantly (*p* = 0.003).

Possible explanations may include repetitive microtrauma, muscular overuse, and psychosocial stress associated with competitive activity. However, this finding should be interpreted with caution. Confounding factors such as training intensity, bruxism, and unmeasured psychological stress may contribute to this association. Further targeted studies are warranted to clarify the causal relationship between athletic activity and TMD risk.

As expected, mean Helkimo scores were substantially higher in the TMD group (13.2 ± 4.8) versus controls (1.8 ± 1.6; *p* < 0.001, d = 2.82).

Table 4 highlights the significant differences in subjective pain perception and functional limitation. The median NPRS scores for both joint and cervical regions were substantially higher in the TMD group (*p* < 0.001), confirming the clinical symptomatology detected by mHI and DC/TMD. The increase in lateral and vertical restriction scores further supports the presence of functional impairment in mandibular motion, reinforcing the clinical utility of mHI as a rapid identifier of symptomatic dysfunction.

Reliability indicators confirmed the strong reproducibility of the mHI. Inter-examiner reliability was excellent (ICC = 0.87), and intra-examiner repeatability after 7 days was even higher (ICC = 0.91). Item-level κ was 0.65, indicating good-to-very good agreement.

Receiver Operating Characteristic (ROC) curves comparing the diagnostic performance of the modified Helkimo Index (mHI) and the DC/TMD protocol in detecting temporomandibular disorders (TMD) (Figure 1). The area under the curve (AUC) was 0.89 for mHI and 0.95 for DC/TMD, indicating excellent diagnostic accuracy for both tools. The diagonal dashed line represents a random classifier (AUC = 0.50).

Mean reliability coefficients (ICC/κ) with 95% confidence intervals for the modified Helkimo Index across three domains: inter-examiner agreement, intra-examiner repeatability, and item-level concordance. Dashed gray line indicates the “good” reliability threshold (ICC/κ = 0.75); dashed black line marks the “excellent” threshold (ICC/κ = 0.90) (Figure 2).

Diagnostic accuracy and agreement between mHI and DC/TMD are summarized in Table 5.

Both instruments demonstrated strong discriminatory ability, but mHI required significantly less application time (5–10 min vs. 16–20 min).

In the comparative accuracy plot (Figure 3), the mHI achieved a sensitivity of 0.86, specificity of 0.84, positive predictive value (PPV) of 0.88, and negative predictive value (NPV) of 0.86. In contrast, DC/TMD reached 0.91, 0.93, 0.91, and 0.89, respectively. Despite a specificity gap of 0.09, the overall diagnostic performance remains comparable, supporting the clinical viability of the Helkimo Index as a lower-complexity alternative to DC/TMD.

Table 6 presents the GEE-logit models fitted separately to the mHI and the DC/TMD scores for predicting temporomandibular disorder (TMD) status. A one-point increase in score raises the adjusted odds of TMD by 45% for mHI (OR = 1.45; 95% CI 1.22–1.73) and by 68% for DC/TMD (OR = 1.68; 95% CI 1.35–2.09), both highly significant (*p* < 0.001). Age and sex show no independent effect, as their confidence intervals cross unity. Examiner intraclass correlation coefficients are low (0.12 for mHI, 0.10 for DC/TMD), indicating minimal rater-related variance. The quasi-likelihood information criterion (QIC) slightly favors the DC/TMD model (215.8 vs. 223.5), yet the difference is clinically negligible. Importantly, the Helkimo assessment requires only ~5–10 min versus 16–20 min for the full DC/TMD protocol, highlighting its operational advantage without sacrificing diagnostic robustness.

The logarithmic curve (Figure 4) illustrates the likelihood ratios for mHI (LR^+^ = 7.82; LR^−^ = 0.16) and DC/TMD (LR^+^ = 10.10; LR^−^ = 0.09). The DC/TMD values exceed the thresholds for strong diagnostic change (LR^+^ > 10; LR^−^ < 0.1), while the mHI still demonstrates a considerable clinical impact, moderate-to-strong in the positive and negative directions. These findings support the utility of the modified Helkimo Index as a rapid triage tool for TMD in outpatient settings with limited resources and reduced personnel.

## 4. Discussion

The primary objective of this study was to validate the modified Helkimo Index (mHI) for the diagnosis of temporomandibular disorders (TMD) in a Romanian patient cohort, using the Diagnostic Criteria for Temporomandibular Disorders (DC/TMD) as the reference standard. Our results indicate that the mHI is a reliable and time-efficient clinical instrument, demonstrating diagnostic performance comparable to that of the DC/TMD protocol. The mHI may serve as a practical tool for initial patient triage and assessment, particularly in resource-limited clinical settings or during large-scale population screening programs. These findings provide strong evidence supporting the accuracy and reproducibility of this simplified clinical tool, highlighting its potential for integration into routine practice where full DC/TMD implementation may be logistically challenging.

This makes it a valuable alternative for initial patient triage and assessment, particularly in outpatient dental clinics where the full DC/TMD protocol may be logistically challenging or impractical due to time constraints.

The demographic analysis (Table 3) showed that the TMD and control groups were comparable in gender distribution and body mass index (BMI), minimizing potential confounding effects from these variables. A noteworthy finding from our demographic analysis was the significant age difference between the groups, with the TMD group being slightly younger (28.0 ± 6.0 years) than the control group (31.0 ± 6.5 years; *p* = 0.02). This is consistent with previous research suggesting a higher prevalence of TMD symptoms in younger individuals, identifying age as a potential risk factor. Lövgren et al. reported a higher prevalence of TMD symptoms in younger age groups, highlighting age as a relevant demographic risk factor [13]. Similarly, in a Dutch adolescent population aged 12–18 years, the prevalence of painful TMD was reported at 21.6%, with logistic regression identifying age as a significant predictor of TMD risk (*p* < 0.05) [22].

We also found a significant association between athletic activity and TMD, with a higher proportion of TMD patients being athletes (60%) compared to healthy controls (48%). This finding aligns with studies by Crincoli et al. and Freiwald et al. [23,24], which suggested that the mechanical stress, repetitive microtrauma, and muscular overuse associated with high-intensity training can contribute to TMD onset and exacerbation in athletes participating in contact sports such as rugby and American football. Specifically, they observed increased rates of arthralgia, masticatory muscle pain, and limitations in mandibular movement, particularly lateral excursions, compared to non-athletic controls. Freiwald et al. also reported a significantly higher risk of TMD in athletic populations compared to the general population [24]. However, it is important to interpret this, as other factors like bruxism or psychological stress were not measured and could be confounding variables.

In addition, our study identified a significantly higher prevalence of TMD among individuals residing in rural areas (60%) compared to those in urban areas (40%), with statistical significance (*p* = 0.003; Φ = 0.280). This observation supports the findings of Montero et al. [3], who demonstrated that sociodemographic factors, especially rural residency, can adversely affect orofacial health. Limited access to specialized dental services, lower health literacy, and delayed diagnosis increased vulnerability among rural populations. These findings underscore the importance of incorporating sociodemographic factors into clinical assessments and public health policies aimed at prevention. Another study reported that rural residents experience a higher incidence of facial pain (46.2% vs. 20.2%; *p* < 0.01) and more frequent occurrences of TMD, disk dislocations, and degenerative joint disorders compared to their urban counterparts [25].

From a diagnostic performance perspective, the modified Helkimo Index demonstrated a sensitivity of 86% and a specificity of 84%, values closely aligned with those of the DC/TMD protocol (sensitivity 91%, specificity 93%). This similarity confirms the mHI’s capacity to discriminate between patients with and without TMD. The strong correlation between mHI and DC/TMD scores (r = 0.83) further validates the mHI as a reliable clinical alternative. In support of these results, Alonso-Royo et al. [17] reported a sensitivity of 86.7% and specificity of 68.1% for the Helkimo Clinical Dysfunction Index compared to the DC/TMD in a clinical sample. These findings reinforce the diagnostic value of mHI and advocate for its broader use, especially in low-resource environments.

The area under the curve (AUC) for the mHI was 0.89, confirming its excellent ability to differentiate between TMD and non-TMD cases, a performance that slightly exceeds some previously reported values for the Helkimo score, where the AUC ranged from 0.84 to 0.87 [2,17].

The optimal cutoff point identified (≥9 points), along with positive and negative predictive values and likelihood ratios, supports the validity of the mHI as a reliable, rapid, and effective clinical screening tool for TMD, particularly in resource-constrained settings.

Our results regarding the reproducibility of the mHI revealed excellent consistency, with an inter-examiner intraclass correlation coefficient (ICC) of 0.87 and an intra-examiner ICC of 0.91, surpassing conventional thresholds for reliability. These values align with findings by Alonso Royo et al. [17], who reported ICCs ranging from 0.85 to 0.90 for the Helkimo Clinical Dysfunction Index (HCDI) in TMD assessments. Furthermore, a median kappa value of 0.65 observed in our study suggests a good-to-very-good agreement at the item level, comparable to the weighted kappa coefficients reported by Alonso Royo et al., which ranged from 0.43 to 0.77 [17]. These results demonstrate both temporal and inter-observer stability, reinforcing the utility of the mHI in routine clinical practice and epidemiological research without compromising diagnostic rigor.

Multivariate analysis using Generalized Estimating Equations (GEE) highlighted a strong predictive relationship between mHI scores and the likelihood of a positive TMD diagnosis. Every additional point on the Helkimo Index was associated with a 45% increase in adjusted odds. This significant predictive association confirms the internal validity of the mHI. Moreover, the absence of independent effects of age or sex emphasizes the index’s symptom-specific nature, indicating that it accurately reflects dysfunction severity regardless of demographic variables. These findings are consistent with those of Yarasca-Berrocal et al. [15], who compared the mHI to the short-form Fonseca Anamnestic Index (SFAI) and reported an AUC of 0.854 for mHI, with a sensitivity of 89.7% and specificity of 77.6%. This reinforces the high discriminative power of the Helkimo score in detecting TMD cases, further validating its utility for rapid clinical screening and epidemiological studies. Hence, the advanced statistical modeling (GEE) and external validation comparisons advocate for the mHI as a pragmatic alternative to more complex diagnostic protocols like the DC/TMD, especially in environments with limited resources.

The optimal administration time of a clinical tool is essential for its efficient implementation in dental outpatient settings. Our findings show that the mHI can be completed in 5–10 min, substantially less than the 16–20 min required for the full DC/TMD protocol. This time difference is critical, as it reduces both patient burden and examination duration without compromising diagnostic accuracy. Moreover, validation studies, including Alonso Royo et al. [17], affirm that the HCDI is a “simple and rapid test with adequate clinimetric properties.” While the brief version of the DC/TMD (bDC/TMD) provides a screening protocol lasting approximately 10 min, the complete Axis II interview can add 10–15 min [26]. A recent study validating the Krogh-Poulsen test, a fast, composite screening tool, reported an AUC of 0.93 and a short administration time, making it suitable for primary care settings without specialized staff [27]. Put together, these data suggest that the mHI offers an ideal balance between speed and accuracy, making it a viable solution for early-stage TMD diagnosis with efficient resource use.

Beyond the Romanian context, several studies worldwide have applied or validated the modified Helkimo Index, underscoring its broader clinical relevance. For example, Alonso-Royo et al. (Spain) validated the Helkimo Clinical Dysfunction Index for TMD diagnosis [17]; Yarasca-Berrocal et al. (Peru) compared the predictive accuracy of the mHI with the short-form Fonseca Index [15]; Rani et al. (India) analyzed the Helkimo Index in dental students [14]; and Crincoli et al. (Italy) explored TMD prevalence in athletes using clinical indexes, including Helkimo-based assessments [23]. Integrating these findings situates our study within the broader international research landscape and highlights the global applicability of the mHI.

### 4.1. Strengths and Limitations

This study presents a rigorous clinical validation of the modified Helkimo Index (mHI) in a Romanian patient sample, using the DC/TMD protocol as the reference standard. The sample size was sufficient to support robust statistical analyses, and the comparison groups were carefully selected to ensure demographic comparability. The mHI demonstrated excellent diagnostic performance, with 86% sensitivity, 84% specificity, and a strong correlation (r = 0.83) with DC/TMD scores. Reliability analyses revealed excellent inter- and intra-examiner agreement (ICC = 0.87 and 0.91, respectively), confirming the tool’s stability and reproducibility. In addition, its short administration time (5–10 min) offers a significant operational advantage over more complex protocols, supporting its use in clinical environments with limited resources.

Nonetheless, several limitations must be acknowledged. Although the sample size was adequate for statistical analysis, the recruitment of participants from a single hospital and within a relatively homogenous age group restricts the external validity of the findings. Moreover, the study was conducted in a specialized clinical setting in Romania, which may limit the applicability of the results to broader populations and routine dental practice. Psychosocial variables such as bruxism, stress, and anxiety were not directly assessed and may act as unmeasured confounders influencing temporomandibular disorder expression. To strengthen generalizability, future multi-center investigations should include more diverse socio-demographic cohorts, incorporate psychosocial risk factors, and extend validation across different clinical contexts and cultural environments.

### 4.2. Clinical Implications

These findings have significant clinical implications. Validation of the modified Helkimo Index in a Romanian sample provides dental practitioners and researchers with a simple, fast, and effective tool for assessing temporomandibular disorders, complementing the DC/TMD protocol. Implementing this index can improve access to timely and accurate diagnoses, reducing the time and resources required while facilitating early detection and optimal management of this complex condition.

## 5. Conclusions

Our study demonstrates that the modified Helkimo Index (mHI) is a valid, reliable, and operationally efficient clinical alternative to the DC/TMD protocol for diagnosing temporomandibular disorders. Its high performance in terms of sensitivity, specificity, and reproducibility, combined with its short administration time, supports the broad implementation of the mHI in general dental practice and resource-limited settings. These results validate the mHI as a pragmatic tool with genuine potential to optimize triage and the early diagnosis of TMD in diverse clinical contexts.

## Figures and Tables

**Figure 1 diagnostics-15-02347-f001:**
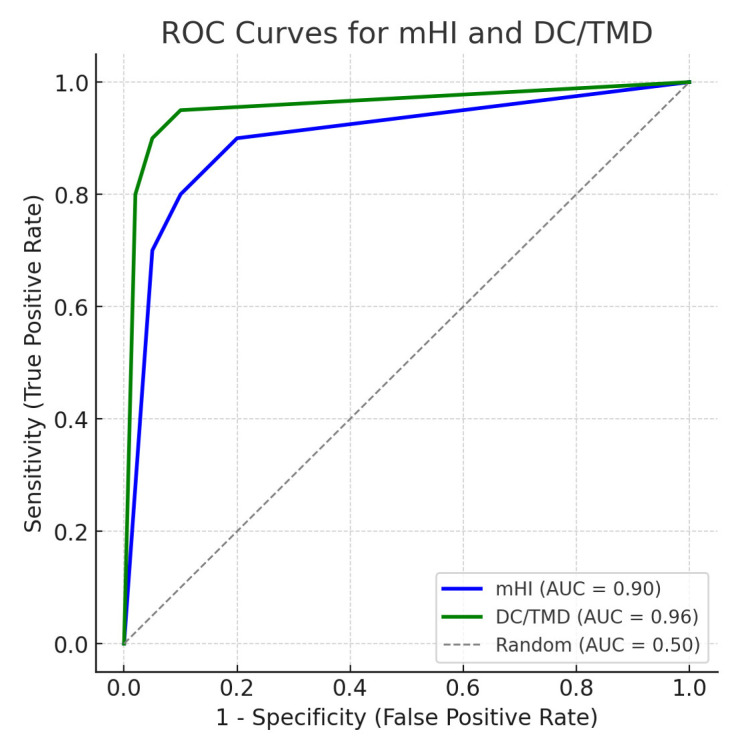
Receiver Operating Characteristics (ROC) curves for the diagnostic performance of mHI and DC/TMD.

**Figure 2 diagnostics-15-02347-f002:**
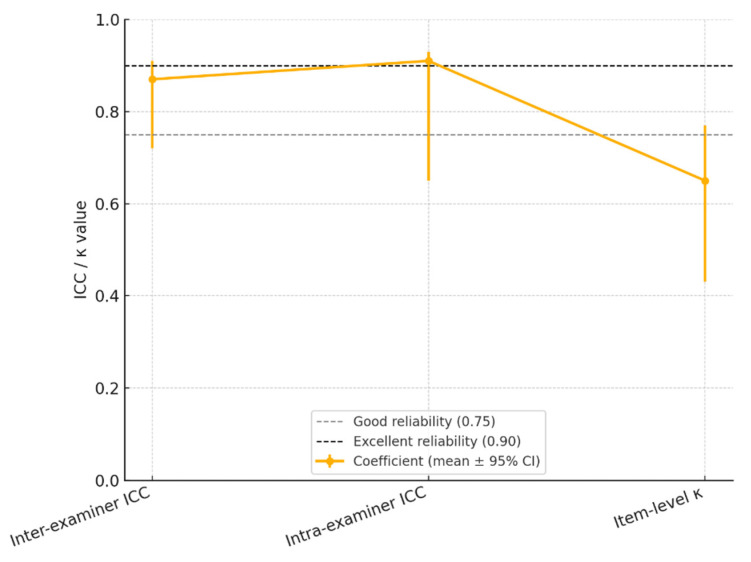
Reliability Landscape of the Modified Helkimo Clinical Index: Inter-Examiner, Intra-Examiner, and Item-Level Agreement (ICC/κ with 95% CI.

**Figure 3 diagnostics-15-02347-f003:**
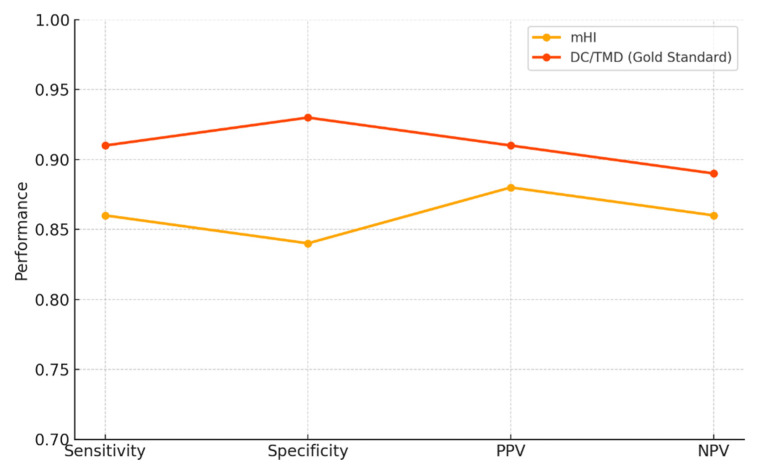
Head-to-head accuracy profile of the mHI versus DC/TMD.

**Figure 4 diagnostics-15-02347-f004:**
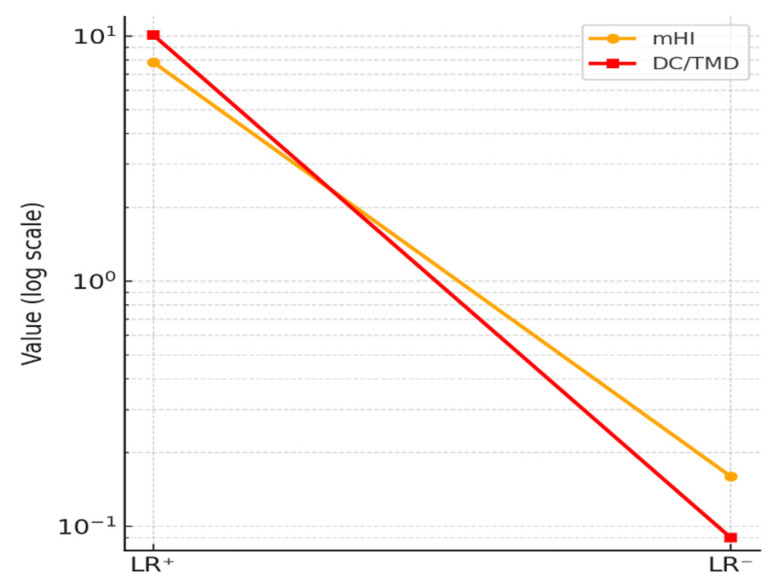
Likelihood ratios (log scale).

**Table 1 diagnostics-15-02347-t001:** Assessment of TDM using the Helkimo Index scale.

Parameter	Description	Evaluation	Scoring Scale
A. Limitation of mandibular range of motion	Assessment of maximum opening, lateral movements, and mandibular protrusion	Direct clinical measurements	- Vertical opening: ≥40 mm—0 points (normal) 30–39 mm—1 point (mild limitation) <30 mm—5 points (severe limitation) - Lateral movements and protrusion: ≥7 mm—0 points 4–6 mm—1 point 0–3 mm—5 points
B. Alterations in joint function	Observation during mandibular opening/closing by palpation and auscultation	Detection of joint sounds, locking, or dislocation	- No deviation or sounds—0 points - oint sounds or mandibular deviation—1 point - Locking or dislocation (with or without sounds)—5 points
C. Pain during movement	Self-reported pain during mandibular movements		- No pain—0 points - Pain during a single movement—1 point - Pain during two or more movements—5 points
D. Muscle pain	Palpation or functional manipulation of masticatory muscles	Identification of painful areas	- No pain on palpation—0 points - Pain in 3 zones—1 point - Pain in 4 or more zones—5 points
E. TMJ pain	Palpation of the periauricular area and external auditory canal	Evaluation of pain on palpation	- No spontaneous or palpation-induced pain—0 points - Pain on uni/bilateral periauricular palpation—1 point - Pain on palpation of both the external auditory canal and periauricular area—5 points

**Table 2 diagnostics-15-02347-t002:** Severity classification for TMD based on modified Helkimo Index.

Score Range	Severity Level
0	Absence of temporomandibular disorder (TMD)
1–9	Mild form of TMD
10–19	Moderate manifestation of TMD
20–25	Severe stage of TMD

**Table 3 diagnostics-15-02347-t003:** Baseline Demographic and Clinical Characteristics.

Caracteristic	Total (*n* = 164)	TMD (*n* = 82)	Control (*n* = 82)	Test *	* p * -Value	Effect Size **
Age, years—mean ± SD	29.5 ± 6.30	28.0 ± 6.00	31.0 ± 6.50	t (Welch)	0.020	d = 0.490
Sex, F/M, n (%)	86/78 (52%/48%)	42/40 (51%/49%)	46/36 (56%/44%)	χ^2^	0.350	Φ = 0.070
BMI, kg/m^2^—mean ± SD	24.50 ± 3.20	24.70 ± 3.40	24.30 ± 3.10	t	0.370	d = 0.120
Athletes—yes/no, n (%)	88/76 (54%/46%)	49/33 (60%/40%)	39/43 (48%/52%)	χ^2^	0.004	Φ = 0.270
Residence—rural/urban, n (%)	76/88 (46%/54%)	49/33 (60%/40%)	27/55 (33%/67%)	χ^2^	0.003	Φ = 0.280
Helkimo Score—mean ± SD	7.50 ± 7.10	13.20 ± 4.80	1.80 ± 1.60	t	<0.001	d = 2.820

* The applied statistical tests depend on the actual distribution of each variable (Kolmogorov–Smirnov/Shapiro–Wilk tests for normality). ** Effect size: Cohen’s d (normally distributed continuous variables), r coefficient (Mann–Whitney), or Phi coefficient (Φ) for categorical variables.

**Table 4 diagnostics-15-02347-t004:** NPRS Pain Levels and Functional Scores.

Domain	*n* Patients (*n* Examiners)	Examiner 1 (Min–Max; Mean ± SD)	Examiner 2 (Min–Max; Mean ± SD)	ICC/κ (95% CI)	SEM	MDC95	Classification *
1. Inter-examiner	164 (2)	0–25; 7.4 ± 6.9	0–24; 7.6 ± 7.0	0.87 (0.72–0.91)	1.4 pts	4.0 pts	> 0.90
2. Intra-examiner **	35 (1)	1–23; 8.1 ± 5.8	1–22; 8.0 ± 5.6	0.91 (0.65–0.93)	1.1 pts	3.1 pts	> 0.90
3. Item-level κ (median)	164 (2)	n/a	n/a	0.65 (0.43–0.77)	n/a	n/a	0.75–0.90

* Classification based on commonly accepted ICC/κ interpretive thresholds [21]: Poor (< 0.50), Moderate (0.50–0.75), Good (0.75–0.90), Excellent (> 0.90). ** Re-test after 7 days on a subsample of 35 patients, with no interventions between evaluations.

**Table 5 diagnostics-15-02347-t005:** Consolidated Diagnostic Accuracy and Agreement: mHI vs. DC/TMD.

Parameter	mHI	95% CI	DC/TMD	95% CI	Interpretation
Sensitivity	0.86	0.78–0.92	0.91	0.83–0.95	Both excellent
Specificity	0.84	0.81–0.94	0.93	0.85–0.97	DC/TMD slightly higher
AUC	0.89	—	0.95	—	Very strong discrimination
PPV	0.88	0.81–0.93	0.91	0.83–0.96	Comparable
NPV	0.86	0.78–0.92	0.89	0.80–0.94	Comparable
LR^+^	7.82	4.70–13.0	10.1	5.8–26.4	Stronger for DC/TMD
LR^−^	0.16	0.10–0.27	0.09	0.05–0.18	Stronger for DC/TMD
Correlation (r)	0.83	*p* < 0.001	—	—	High inter-method agreement
ICC inter-method	0.87	0.72–0.91	0.87	0.72–0.91	Consistent agreement

**Table 6 diagnostics-15-02347-t006:** GEE-logit model − TMD diagnosis (mHI vs. DC/TMD).

Variable	mHI − OR [95% CI]	*p*	DC/TMD − OR [95% CI]	*p*	Interpretation
Score (per 1 point)	1.45 [1.22–1.73]	<0.001	1.68 [1.35–2.09]	<0.001	↑ risk ~45–68% per point
Age (years)	1.05 [0.99–1.11]	0.100	1.03 [0.97–1.09]	0.340	not significant
Sex (female)	1.30 [0.74–2.30]	0.360	1.22 [0.65–2.30]	0.540	not significant
Examination duration *	≈5–10 min (rapid eval.)	n/a	≈16–20 min (full protocol)	n/a	n/a

* Duration refers to the time required to complete the full clinical examination (excluding the auxiliary questionnaire).

## Data Availability

Datasets are available from the authors upon reasonable request.

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
