# Peer review of "Validation of the Modified Helkimo Clinical Index for Diagnosing Temporomandibular Disorders in a Romanian Patient Sample"

_diagnostics, 2025, doi:10.3390/diagnostics15182347_

Round 1
Reviewer 1 Report
Comments and Suggestions for Authors
First of all, I would like to commend the authors for addressing an important and clinically relevant topic. The manuscript is well-structured, the methodology is rigorous, and the statistical analysis is sound. My comments below are intended to further strengthen the clarity and impact of the work.
Abstract: Please consider condensing it to emphasize the main numerical outcomes (sensitivity, specificity, ICC, AUC, and application time) and the overall conclusion, while avoiding too much methodological detail.
Introduction:
- Some sections repeat information (e.g., history of the Helkimo Index) and could be streamlined.
- Please highlight more clearly what is novel in this study compared to prior validations of the Helkimo Index
Methodology:
- Please clarify whether examiners underwent formal calibration/training before starting the study, as this affects reproducibility.
- Exclusion of systemic factors (bruxism, psychological stress, depression) is not fully discussed. Since these are relevant to TMD, please justify their omission or clarify how they were accounted for.
Results:
- Some tables (particularly Tables 5, 6, and 8) present overlapping information. Merging them into a consolidated table may improve readability.
- Figures are informative but a bit dense. Simplifying or merging (e.g., ROC + AUC comparison in one figure) would make the results more accessible to general readers.
- The finding of higher TMD prevalence in athletes is interesting. Please expand discussion or provide possible explanations (e.g., trauma, muscular overuse, stress), and also caution about confounding factors.
Discussion:
- The discussion is comprehensive but occasionally repetitive. Consider summarizing key points more concisely.
- Please expand the comparison with other validated screening tools to highlight the relative advantage of the mHI.
- The rural vs. urban difference is noteworthy, but causality cannot be established. Please temper the interpretation and acknowledge potential confounding factors such as socioeconomic status or healthcare access.
Best Wishes.
Reviewer 2 Report
Comments and Suggestions for Authors
Dear Authors
I would like to congratulate you for achieving this interesting, well-written study. I have the following remarks that need clarification:
1- Did you calculate the sample size?
2- Why did you recruit your patients from one hospital? Did this affect the generalizability of the findings?
3- What was your explanation regarding the athletes are more susceptible to TMD?
4- In the introduction you mentioned that "DC/TMD protocol may be impractical", why?
5- The introduction lacks the comprehensive information about the modification of the Helkimo Clinical Index for Diagnosing Temporomandibular Disorders?
6- The limitations need expansion.
7- Did you use Welch’s t-test for all parameters?
8- You mention the abbreviation GEE but you did not introduce it?
Regards
Reviewer 3 Report
Comments and Suggestions for Authors
The study is focused on assessing the diagnostic validity and reliability of the mHI in a Romanian patient cohort. This manuscript is well-written and is very interesting for the journal´s readers. I have only a few recommendations.
Abstract: Please revise the abstract and mention the objective in the first section, not in the methods.
Keywords: Please review that the keywords are MeSH terms, this help to the researchers to classify the manuscript
Background- Introduction: Adequate for the research purposes
Methods: Please review that the manuscript accomplishes the STROBE guidelines for observational studies and STARD2015: An Updated List of Essential Items for Reporting Diagnostic Accuracy Studies. It is mandatory that the authors provided the files containing the verification list.
Results: Adequate for the study purposes .
Discussion: Adequate for the study purposes. I would like to know if there are more studies around the world that uses the mHI.
References: I would like to know if there are more studies around the world that uses the mHI.
Tables and figures: Adequate for the study purposes.
Reviewer 4 Report
Comments and Suggestions for Authors
The article entitled „Validation of the Modified Helkimo Clinical Index for Diagnosing Temporomandibular Disorders in a Romanian Patient Sample” analyze an interesting concept on the clinical validation of the mHI as a diagnostic tool for TMD. The manuscript is well written and the research protocol is adequately conceived and implemented. However, I have some recommendations for the improvement of the manuscript:
- What is the novelty of the current study? Previous research on mHI have already been published.
- The last paragraph in the introduction should be relocated in the discussion section: „The mHI is proposed as a practical solution for initial patient triage and assessment, particularly in low-resource clinics or during population-level screening programs. Our findings provide robust evidence supporting the accuracy and reproducibility of this simplified clinical tool, advocating for its integration into routine practice where full implementation of the DC/TMD may be logistically challenging or impractical”.
- Sample size calculation should be performed and added in the text by the authors. Indicate minimum sample size.
